# Metabolic Effects of Recurrent Genetic Aberrations in Multiple Myeloma

**DOI:** 10.3390/cancers13030396

**Published:** 2021-01-21

**Authors:** Timon A. Bloedjes, Guus de Wilde, Jeroen E. J. Guikema

**Affiliations:** 1Department of Pathology, Amsterdam University Medical Centers, Location AMC, University of Amsterdam, 1105 AZ Amsterdam, The Netherlands; t.a.bloedjes@amsterdamumc.nl (T.A.B.); g.dewilde@amsterdamumc.nl (G.d.W.); 2Lymphoma and Myeloma Center Amsterdam (LYMMCARE), 1105 AZ Amsterdam, The Netherlands

**Keywords:** multiple myeloma, cancer metabolism, MYC, cyclin D1, FGFR3, MMSET, MAF

## Abstract

**Simple Summary:**

Cancer is associated with metabolic changes related to increased cell proliferation and growth. These cancer-related metabolic features are largely dictated by specific oncogenes that are activated by chromosomal aberrations and epigenetic alterations in cancer cells. Multiple myeloma is an incurable plasma cell malignancy, which is characterized by recurrent chromosomal aberrations that drive the expression of established oncogenes such as MYC, Cyclin D1, FGFR3/MMSET and MAF/MAFB. In this review, we discuss the specific metabolic features of multiple myeloma plasma cells, and focus on the metabolic consequences of recurrent chromosomal aberrations, thereby providing an outline for the metabolic alterations that characterize multiple myeloma.

**Abstract:**

Oncogene activation and malignant transformation exerts energetic, biosynthetic and redox demands on cancer cells due to increased proliferation, cell growth and tumor microenvironment adaptation. As such, altered metabolism is a hallmark of cancer, which is characterized by the reprogramming of multiple metabolic pathways. Multiple myeloma (MM) is a genetically heterogeneous disease that arises from terminally differentiated B cells. MM is characterized by reciprocal chromosomal translocations that often involve the immunoglobulin loci and a restricted set of partner loci, and complex chromosomal rearrangements that are associated with disease progression. Recurrent chromosomal aberrations in MM result in the aberrant expression of MYC, cyclin D1, FGFR3/MMSET and MAF/MAFB. In recent years, the intricate mechanisms that drive cancer cell metabolism and the many metabolic functions of the aforementioned MM-associated oncogenes have been investigated. Here, we discuss the metabolic consequences of recurrent chromosomal translocations in MM and provide a framework for the identification of metabolic changes that characterize MM cells.

## 1. Introduction

Multiple myeloma (MM) is a neoplastic disease that manifests itself by the accumulation of clonal plasma cells in the bone marrow. MM has a yearly global incidence of about ~5/100,000 individuals, and mainly affects the elderly. Despite recent clinical improvements due to the development of novel drugs, the outcome for MM patients remains dismal. The stepwise malignant transformation of plasma cells leading up to MM is accompanied by genetic and epigenetic changes that have been well studied over the recent years (reviewed in [1,2,3,4]). Additionally, it is becoming increasingly evident that cancer cells display altered metabolic features related to increased demands and rewired regulation of metabolic genes, often caused by oncogene activation. MM plasma cells originate from post-germinal center B cells and are characterized by a number of recurrent genetic aberrations. The presence of chromosomal aberrations that involve the immunoglobulin loci and a restricted set of partner chromosomes/loci constitutes an almost universal event in MM [5,6]. Full-blown MM may be preceded by a premalignant condition termed monoclonal gammopathy of undetermined significance (MGUS) and an asymptomatic phase known as smoldering MM (sMM). Importantly, many of the recurrent chromosomal aberrations can already be detected in the premalignant and asymptomatic conditions, indicating that these are important drivers of malignant transformation in MM [7]. The most recurrently encountered structural genetic abnormalities in MM affect well-known cancer-associated genes such as *MYC* and cyclin D1 (*CCND1*), that supposedly primarily drive malignant proliferation. In addition, translocations involving the fibroblast growth factor receptor 3 (*FGFR3*), the histone methyltransferase multiple myeloma SET domain (*MMSET*) and the musculoaponeurotic fibrosarcoma (*MAF*) oncogene homolog transcription factors are often found. The biological sequelae of these latter translocations are complex and less well understood. The oncogenic consequences of the genes dysregulated by recurrent chromosomal translocations in MM have been studied extensively in recent years; however, their impact on cellular metabolism have received relatively little attention. In this review, we discuss the metabolic characteristics of normal plasma and MM plasma cells, with a particular focus on the metabolic consequences of the dysregulated MM-associated oncogenes CCND1, MYC, FGFR3, MMSET and MAF.

### 1.1. Metabolic Characteristics of Plasma Cells

Plasma cells are highly specialized cells that play a crucial role in humoral immunity by producing and secreting antibodies. Plasma cells generated in a primary immune response develop in a T-cell-independent fashion, are short-lived and secrete mostly low-affinity IgM. These cells and antibodies are part of a first line of immune defense that precedes the emergence of higher-affinity antibodies and long-lived plasma cells, which are generated in a T-cell dependent germinal center (GC) response. Long-lived plasma cells migrate to the bone marrow where they reside to provide lifelong protection by constitutive antibody secretion. It must be noted that the heterogeneity in plasma cells based on longevity is not strict and many characteristics between short-lived and long-lived plasma cells overlap [8]. Regardless, the specialized function of plasma cells to produce and secrete antibodies, and their limited proliferative capacity, poses specific demands on nutrient uptake and synthesis of biomolecules. Plasma cell differentiation initiates with the activation of naïve B cells, which results in the increase of glucose uptake, driving glycolysis and oxidative phosphorylation (OXPHOS) [9]. Antigen-receptor engagement activates the phosphatidylinositol 3-kinase (PI3K) and protein kinase B (AKT) pathway that is of pivotal importance for the orchestration of metabolic changes in B cells [9,10,11]. AKT is known to increases glucose uptake and glycolysis, while inhibiting fatty acid oxidation. At the same time, it increases lipid synthesis, activity of the pentose phosphate pathway (PPP), glutaminolysis and nucleotide synthesis and there is evidence that AKT promotes mitochondrial respiration [12,13,14,15]. In the GC, antigen-activated B cells undergo vigorous proliferation and diversify their antibody repertoire by somatic hypermutation (SHM) of their antibody genes, which requires the expression of activation-induced cytidine deaminase (AID) [16,17]. AID is a B-cell specific enzyme that deaminates cytosines in immunoglobulin gene segments that code for the antigen-binding domains of antibodies. Cytosine deamination results in genomic uracils, which are processed in an error-prone fashion by the base excision repair and mismatch repair pathways, resulting in mutations, thereby creating antibody variants with altered affinity for antigen. Ensuing T-cell dependent positive selection in the GC based on antigen binding results in affinity maturation and the generation of long-lived plasma cells that secrete high affinity antibodies. Next to SHM, AID is also crucial for class switch recombination (CSR) as processing of AID-instigated genomic uracils in the switch (S) regions of the immunoglobulin heavy chain (*IGH)* locus primarily results in the formation of DNA double-strand breaks (DSB) that are required for CSR [18,19]. Of interest, AID has been firmly implicated in the generation of recurrent chromosomal translocations and the acquisition of driver mutations in MM, implicating a GC origin for MM [20,21]. Clonal B-cell expansion in the GC requires the mammalian target of rapamycin complex 1 (mTORC1) and the glycogen synthase kinase 3 (GSK3), which drive glycolysis and mitochondrial biogenesis [22,23]. The relative sparsity of oxygen in the GC microenvironment would suggest that GC B cells primarily rely on glycolysis [24,25], since OXPHOS requires oxygen. However, a recent study suggests that highly proliferative GC B cells are not glycolytic but instead use fatty acid oxidation for OXPHOS [26]. The latter results would be in line with a rather modest glucose uptake that was reported for GC B cells [22]. Next to utilization of fatty acids for energetic demands, activated B cells evidently also synthesize fatty acids to prepare for the expansion of the endoplasmatic reticulum (ER) during plasma cell differentiation [27]. Antibody production requires an extensive ER machinery, associated with increased demands of fatty acid and amino acids. Furthermore, high level antibody production saturates the protein folding capacity of the ER, resulting in an ER stress response in plasma cells [28,29]. Protein synthesis is regulated by the nutrient sensor mTORC1 that is activated by amino acids. Rapamycin treatment abrogated plasma cell differentiation in mice, indicating that mTORC1 is of crucial importance for the generation of plasma cells. Moreover, antibody production was decreased by rapamycin treatment, whereas the frequency of long-lived plasma cells in the bone marrow was not affected [30]. These results indicate that mTORC1 primarily functions to regulate antibody biosynthesis in plasma cells. The metabolic reprogramming that takes place in plasma cells is transcriptionally dictated by the transcription factors B lymphocyte-induced maturation protein-1 (BLIMP1) and the interferon regulatory factor 4 (IRF4), which silence the paired-box 5 (PAX5) transcription factor that is involved in the suppression of glycolysis [31]. Using ex vivo lipopolysaccharide (LPS)-induced differentiation of murine B cells it was demonstrated that OXPHOS is increased upon T-cell-independent plasma cell differentiation. However, LPS-induced plasma cells still rely on glycolysis, likely because it produces pyruvate that fuels OXPHOS. BLIMP1 was required for this rise in OXPHOS [32], perhaps by preventing PAX5-mediated suppression of glycolysis. Pyruvate derived from glycolysis also crucially fuels OXPHOS in long-lived plasma cells in vivo, as shown by the specific loss of these cells in mice with a deletion of the mitochondrial pyruvate carrier 2 (*MPC2*) [33]. Additionally, these results suggest that glutaminolysis can not compensate to maintain OXPHOS when glycolysis is hampered in plasma cells. Concomitantly, inhibition of glycolysis by treatment with 2-deoxyglucose (2-DG) impaired LPS-induced plasma cell differentiation, whereas retroviral overexpression of the rate-limiting glycolysis enzyme hexokinase 2 (HK2) drove plasma cell differentiation, accompanied by PAX5 downregulation [34].

The crucial impact of metabolic features on plasma cells was recently demonstrated in a direct comparison of long-lived versus short-lived plasma cells. Single cell transcriptomics of these different plasma cell populations revealed few differences, whereas glucose and amino acid uptake was clearly increased in long-lived plasma cells. These results suggest that lifespan of plasma cells is largely determined by metabolic features and is not transcriptionally regulated. Importantly, these metabolic alterations were not related to the ER stress response, as this was comparable in both plasma cell populations. It was demonstrated that long-lived plasma cells express higher levels of CD98 (*SLC3A2*), a common subunit of several amino acid transporters, which may be at the basis of the metabolic differences between short-lived and long-lived plasma cells [35]. In addition, we speculate that differential signaling characteristics dependent on microenvironmental niche variation and nutrient availability may underlie the functional metabolic differences between long-lived and short-lived plasma cells. In activated B cells, glucose is primarily used for glycolysis; however, in plasma cells, glucose also has a different fate and is generally used for the glycosylation of antibodies, which involves the hexosamine biosynthesis pathway (HBP). This pathway siphons the glycolysis intermediate fructose 6-phosphate off to produce uridine diphosphate-N-acetylglucosamine (UDP-GlcNac) that is used as a glycosylation substrate. The HBP converges with glutamine, fatty acid and nucleotide metabolism, requiring several intermediates derived from these pathways. The X-box binding protein-1 (XBP1) transcription factor is activated by the ER stress response in plasma cells and directly activates the HBP by transcriptional regulation of several rate-limiting enzymes [36]. This illustrates that increased protein production crucially impinges on metabolic pathways in plasma cells. Interestingly, low adenosine triphosphate (ATP) conditions in long-lived plasma cells may force glucose down the glycolysis pathway to generate pyruvate to fuel OXPHOS for energetic demands [33]. These observations indicate that plasma cells display some degree of metabolic plasticity.

### 1.2. Metabolic Characteristics of MM Cells

The activation of oncogenes triggers malignant cellular transformation, which is often associated with the dramatic reprogramming of metabolic pathways in cancer cells. An altered metabolic state is apparent in all types of cancer, including MM, and can be considered as one of the hallmarks of cancer [37]. The marked increase in glucose uptake in cancer cells was described by Otto von Warburg in 1927 [38], a phenomenon that was later dubbed the “Warburg effect”. The prevailing thought was that cancer cells ferment glucose in the presence of oxygen, termed aerobic glycolysis, at the expense of OXPHOS. However, this concept was challenged by the finding that many tumors in vivo exhibit sufficient or even elevated OXPHOS [39,40,41]. MM cells were reported to be particularly reliant on glycolysis, as shown by the specific sensitivity for the glycolysis inhibitor dichloroacetate [42]. Moreover, the glycolysis rate-limiting enzyme HK2 is highly expressed in newly diagnosed MM cases, increases with disease progression and is associated with poor prognosis [43,44,45]. Of interest, the end product of glycolysis, lactate, is typically secreted by cells; however, MM cells also take up exogenous lactate through the monocarboxylate transporter 1 (MCT1) to fuel OXPHOS [46]. Furthermore, lactate dehydrogenase A (LDHA), the enzyme that converts lactate into pyruvate, is highly expressed in MM cells [44]. However, MM cells clearly also utilize OXPHOS, since co-treatment with the OXPHOS inhibitor metformin and the glucose uptake inhibitor ritonavir synergized to induce cell death in MM cells, indicating that OXPHOS may compensate when glycolysis is limiting. Furthermore, when glycolysis was inhibited, MM cells became increasingly dependent on glutamine, demonstrating that under these conditions OXPHOS was primarily fueled by glutaminolysis [47]. MM cells clearly differ from normal plasma cells in this regard [33], and these results underscore that MM cells possess noticeable metabolic plasticity that may be exploited therapeutically.

Transcriptional rewiring caused by oncogene dysregulation in combination with clonal selection towards the effective acquisition and utilization of various nutrients from a relatively nutrient-starved microenvironment, shapes the metabolic features of cancer cells. Not surprisingly, this is also the case for MM cells that typically reside in the sparsely oxygenated bone marrow environment. Next to the increased metabolic demands associated with antibody secretion, MM plasma cells also have specific demands towards oncogene-driven cellular proliferation, which is not the case for normal plasma cells. These increased metabolic demands represent attractive therapeutic targets that are being explored in a clinical setting. In general, MM cells are characterized by the increased metabolism of glucose, glutamine, fatty acid and nucleotides, largely dictated by the activation of specific oncogenes (Figure 1), which we will discuss in the ensuing chapters of this review.

## 2. C-MYC Deregulation in MM

The transcription factor *c-MYC* (MYC) was originally identified as the cellular homolog of the avian retrovirus-encoded *v-MYC* oncogene that is responsible for viral transforming activity [48,49]. MYC forms heterodimers with its ubiquitously expressed binding partner MYC associated protein X (MAX), which is essential for the gene regulatory function of MYC that predominantly impinges on genes involved in cell growth and proliferation. Interestingly, MYC does not appear to function as a classical on/off determinant for gene transcription, but rather as an amplifier of active genes, partly explaining its highly context-dependent transcriptional effects [50,51]. *MYC* is an established oncogene that is deregulated in the majority of human cancers [52]. In solid tumors, the *MYC* locus at 8q24 is often amplified, whereas in hematological cancers, *MYC* dysregulation is often a result of chromosomal translocations. The chromosomal translocation that places the *MYC* gene in close proximity to the immunoglobulin heavy chain (*IGH*) enhancer regions is the hallmark feature of Burkitt lymphoma [53], and is also frequently observed in other mature B-cell lymphomas, most prominently in diffuse large B-cell lymphoma (DLBCL), where it is associated with disease aggressiveness and adverse prognosis [54]. Chromosomal breakpoint analysis has revealed that illegitimate immunoglobulin class switch recombination (CSR) is responsible for majority of the chromosomal translocations involving *MYC* in B-cell lymphomas, as these are often located in the *IGH* switch (S) regions that are targeted by activation-induced cytidine deaminase (AID) during CSR [55]. In MM, structural variants (SV) involving the *MYC* locus are recurrently observed [5]. A recent report identified *MYC* SVs in 42% of newly diagnosed MM, and in 24% of smoldering MM (sMM) cases, whereas these were absent in monoclonal gammopathy of undetermined significance (MGUS) cases. MGUS and sMM are considered as the premalignant and asymptomatic phases of MM, respectively, suggesting that *MYC* SVs are acquired as secondary genetic events. In MM, the *IGH* enhancer regions are involved in approximately one third of the *MYC* SV cases. The *IGH MYC* rearrangements in MM are characterized by chromosomal duplications of the breakpoint regions and often involve other genes in addition to *IGH* and *MYC* [56]. Initially, it was reported that *MYC* aberrations confer adverse prognosis in MM [57,58], which appeared to be the case, particularly for *MYC* translocations involving the immunoglobulin lambda light chain (*IGL*) [59]. However, in a more recent study the association between *MYC* SVs and poor prognosis in MM could not be corroborated [56].

It is important to note that MYC deregulation does not occur in isolation but requires additional oncogenic changes for its tumor-promoting activities. For instance, it was shown recently that the combined expression of an oncogenic NRAS variant (*NRASQ61R*) and a *MYC* transgene in germinal center B cells was sufficient to drive myelomagenesis in a conditional mouse model [60]. Moreover, in non-transformed cells, several checkpoints prevent the deregulation of MYC expression. Acute elevated MYC expression was shown to activate ARF and BIM, resulting in p53-dependent apoptosis and cell cycle arrest [61,62]. The activation of ARF was demonstrated to require the FOXO3 transcription factor, and it was shown that loss of FOXO activity accelerated the development of MYC-driven lymphomas [63]. The AKT kinase is the pivotal negative regulator of FOXO and is constitutively active in MM [64]. Importantly, we have recently demonstrated that activation of FOXO induces cell cycle arrest and cell death of MM cells. In agreement, inhibition of AKT in MM cells decreased MYC expression and activity [65]. MYC-driven lymphomas in experimental mouse models almost uniformly lose ARF or p53 expression [61], and in human Burkitt lymphoma TP53 is lost in a large fraction of the cases [66]. Additionally, the posttranslational regulation of MYC may also be affected by oncogenic events. For instance, MYC protein stability is increased by ERK activation and/or oncogenic KRAS [67,68], which are frequently implicated in MM [69,70]. In MM, *MYC* SVs did not show any significant association with other genetic events, and it appears that activation of MYC, RAS or NFKB are complementary, showing functional redundancy [56]. Strikingly, *MYC* SV did not correlate with MYC expression level nor with proliferation gene expression index in MM, in contrast to most other cancers. These results indicate that the role of MYC in MM might be different from that in other cancers, perhaps it is not primarily involved in proliferation, but rather in the regulation of the cellular metabolism. As the transcriptional consequences of MYC activation are highly context-dependent, the effects of MYC deregulation in cancer cells with regards to cellular metabolism are also highly heterogeneous and depend on the cell of origin and microenvironmental factors.

### 2.1. MYC and Glucose Metabolism in MM

Importantly, MYC is a key player in the regulation of aerobic glycolysis, which is a hallmark metabolic feature of almost all cancers, as mentioned earlier. MYC-induced cellular transformation was shown to be accompanied by increased consumption and utilization of glucose and glutamine [71,72,73]. MYC was demonstrated to directly activate the expression of many genes involved in glycolysis by binding to the E-box sequence (CACTGT) present in the promoters of key glycolysis genes (Figure 2, Table A1). It was shown by chromatin immunoprecipitation (ChIP) that the major rate-limiting enzymes in glycolysis, such as *HK2*, *LDHA* and enolase 1 (*ENO1*), as well as the glucose transporter gene *SLC2A1*, may be directly regulated by MYC, whereas glyceraldehyde-3-phosphate dehydrogenase (*GAPDH*) and triose isomerase (*TPI*) appeared to be indirectly regulated by MYC [74]. Interestingly, p53 directly represses the expression of glycolysis genes [75,76], which may in part underlie the selection for loss of p53 function in *MYC* deregulated cancers that rely heavily on glycolysis for energetic demands and anabolic precursors, including MM. The p53-inducible gene *TIGAR* (TP53-induced glycolysis and apoptosis regulator) is a major determinant downstream of p53, lowering the levels of reactive oxygen species (ROS) and of fructose-2,6-biphosphate, thereby inhibiting glycolysis [77]. Interestingly, TIGAR was shown to be expressed in human MM cell lines, and loss of TIGAR expression resulted in cell death, likely related to increased ROS levels, which could emanate from deregulated MYC expression [78,79]. Whether TIGAR is involved in the regulation of glycolysis in MM cells is unknown.

Of note, MYC has also been described to regulate glycolysis by its specific effect on the alternative splicing of the pyruvate kinase muscle isozyme (*PKM*), increasing the expression of PKM2 isoform that promotes glycolysis, at the expense of PKM1, which primarily drives OXPHOS [80]. Mechanistically, MYC drives the expression of never in mitosis gene A-related kinase 2 (NEK2), which interacts with heterogeneous ribonucleoproteins A1/2 (hnRNPA1/2) that bind *PKM* transcripts and regulates splicing [81]. The expression of PKM2 was found to be associated with MM cell proliferation, and knockdown of PKM2 increased adhesion to stromal cells and drug resistance [82]. Moreover, analysis of several gene expression datasets indicated that *PKM2* expression was associated with poor prognosis in MM, as was also the case for several other key glycolysis genes, including pyruvate dehydrogenase kinase 1 (*PDK1*), monocarboxylate transporter 1 (*MCT1, SLC16A1*) and the extracellular matrix metalloproteinase inducer (*EMMPRIN, CD147*), of which the latter two are involved in lactate efflux, thereby regulating glycolytic flux in MM [83,84] (Figure 2). Recently, several single nucleotide polymorphisms in the *MCT1* and *CD147* genes were identified in MM patients, which were associated with poor prognosis factors such as elevated beta-2-microglobulin and creatinine levels [85].

Additional evidence for the importance of glycolysis in MM is provided by (18)F-fluorodeoxyglucose (FDG) positron emission tomography (PET), which determines glucose uptake by tumor cells in patients. It was demonstrated that (18)-FDG-PET is an important predictor for outcome in MM patients [86], which is likely associated with the expression of glycolysis genes. Moreover, the (18)-FDG PET metabolic tumor burden was shown to be associated with high-risk gene expression in MM [87]; however, a direct comparison between (18)-FDG-PET status, the expression of glycolysis genes, and MYC aberrations has not been reported yet. Notwithstanding, the crucial dependence of MM cells on glycolysis was revealed in several studies that show that inhibition of glycolysis enzymes, or knockdown of glycolysis genes, resulted in apoptosis [42,44,88,89].

The bone marrow in which MM cells reside is a hypoxic microenvironment [90], which may also drive glycolysis, as it does not rely on oxygen consumption. The hypoxia-inducible transcription factors HIF1α and HIF2α are the predominant regulators of the cellular response to hypoxia, acting in concert with other transcriptional coactivators, including MYC. The crosstalk between HIF1/2 and MYC is complex; HIF1α was shown to antagonize MYC by binding and sequestering MAX, and by inducing FOXO3 expression, whereas HIF2α promotes MYC-MAX activity [91]. However, when MYC is overexpressed it can bypass HIF1α mediated repression, and instead cooperate to drive the expression of *PDK1* and *HK2* [92]. It was found that MYC deregulation in MM resulted in the hypoxia-independent constitutive activation of HIF1α, driving the expression of vascular endothelial growth factor (*VEGF*) and angiogenesis [93], which could perhaps contribute to increased metabolic flexibility of MM cells due to increased oxygenation of the BM microenvironment, thereby making the tumor cells less reliant on glycolysis (Figure 1).

MYC-driven glycolysis may also critically contribute to other metabolic pathways by providing crucial intermediates. For instance, the glycolysis intermediate 3-phosphoglycerate is also used to synthesize serine, which is a major one-carbon source for methylation of DNA, RNA and proteins, and thus involved in epigenetic regulation [94]. Furthermore, oncogenic MYC increases the expression of multiple enzymes involved in serine biosynthesis and glutathione production, which counteracts oxidative stress [95]. In addition, MYC was shown to divert glucose to the PPP by upregulating the expression of enzymes of this pathway [96]. The PPP uses the glycolysis intermediate glucose-6-phosphate, yielding ribose-5-phosphate, which is an important precursor for de novo nucleotide synthesis and thus of major importance for dividing cells (Figure 2). A recent report indicated that the long noncoding RNA (lncRNA) protein disulfide isomerase family A pseudogene 1 (*PDIA3P*), which is highly expressed in MM, interacts with MYC and enhances the transactivation of glucose-6-phosphate dehydrogenase (*G6PD*) promoter, upregulating this rate-limiting PPP enzyme [97]. Moreover, pyruvate is the product of glycolysis that enters the tricarboxylic acid (TCA) cycle and is oxidized to generate reduced electron carriers such as nicotinamide adenine dinucleotide (NADH) and flavin adenine dinucleotide (FADH), which donate electrons for the production of ATP during OXPHOS in the mitochondria. In the presence of glucose and oxygen, MYC was shown to drive the TCA cycle by ramping up pyruvate production [98]. However, it appears that MYC predominantly impacts on the TCA cycle by upregulating glutaminolysis [72] (Figure 2, Table A1).

### 2.2. MYC and Glutaminolysis in MM

Early work on the metabolic profiling of mouse myeloma cells has revealed that at steady state primarily glucose and glutamine are consumed, where the rate of glutamine consumption was highest under limiting glucose conditions [99]. Furthermore, in vitro growth of MM cells was severely blunted by the depletion of glutamine [100,101], suggesting that glutaminolysis is of essential importance for energetic demands and anaplerotic reactions. In agreement, it was demonstrated that MM patients have significantly lower serum glutamine levels compared to healthy controls, whereas serum glutamine levels normalized upon achieving complete remission after treatment [102]. MYC plays a central role in the regulation of glutaminolysis at several levels (Figure 2, Table A1). Similar to glycolysis, MYC drives the expression of glutaminolytic genes as shown by expression profiling and ChIP-seq analysis of the Burkitt lymphoma cell line P493-6 that harbors an inducible MYC expression system [51]. The glutamine transporters alanine-serine-cysteine transporter 2 (ASCT2, encoded by the *SLC1A5* gene) and sodium-coupled neutral amino acid transporter 5 (SNAT5, encoded by the *SLC38A5* gene) are under direct transcriptional control of MYC [72]. Moreover, MYC enhances the expression of glutaminase 1 (*GLS1*) by repressing the expression of the microRNA miR-23a/b, which inhibits *GLS1* translation [103]. It was shown that *ASCT2* is overexpressed in MM compared to healthy donors, sMM and MGUS, with highest expression in secondary plasma cell leukemia and MM cell lines representing aggressive end-stage disease, whereas the expression of *SNAT5* and *GLS1* was comparable in MGUS, sMM and MM [104]. In agreement, miR-23a/b was found to be downregulated in MM, and enforced expression of miR-23a/b in MM cell lines suppressed proliferation and survival [105]. Whether this growth suppressive effect is related to its impact on GLS1 expression has not been studied. MM cells primarily depend on ASCT2 for glutamine transport, and shRNA-mediated *ASCT2* knockdown impaired growth of MM cell lines in vitro and in an in vivo mouse xenograft model. In addition, the GLS1 inhibitors CB-839 and BPTES were shown to induce cell death in MM cell lines [104].

Glutamine can also be synthesized de novo from glutamate and ammonia by a reverse reaction catalyzed by glutamine synthetase (GS). Next to GLS1, MYC is also involved in the regulation of GS by transcriptional activation of the base excision repair protein thymine DNA glycosylase (TDG), which promotes active demethylation of the *GS* promoter [106]. Glutamine synthesis primarily takes place in the cytoplasm and contributes to the production of cytoplasmic proteins and nucleotides, whereas GLS1 is found in the mitochondria. Interestingly, GS was not detected in MM cell lines, even when glutamine is depleted, suggesting that MM cells are incapable of performing de novo glutamine synthesis and rely on glutamine uptake for anabolism. It was reported that the *TDG* gene is aberrantly methylated in human MM cell lines [107], which may underlie the inability of MM cells to perform glutamine synthesis, and perhaps in part explain the glutamine dependence of MM cells, but this remains untested. Ammonia is the toxic byproduct of glutamine breakdown. Conversely, ammonia is consumed during glutamine synthesis. In support of the notion that glutamine synthesis is impaired in MM cells, it was demonstrated that MM cells produce ammonia in the presence of glutamine [104,108]. In agreement, several studies report on the occurrence of hyperammonemia in MM patients, which causes serious clinical complications such as encephalopathy [109,110,111]. Whether this phenomenon is more prominent in MM cells that harbor *MYC* SVs is unknown.

Glutamine is converted into glutamate by GLS1 and subsequently converted into the TCA intermediate alpha-ketoglutarate (αKG) by glutamate dehydrogenase (GDH). Alternatively, transaminases such as glutamic oxaloacetic transaminase 1 and 2 (GOT1, GOT2) may convert glutamate in αKG. In mammary epithelial cells it was demonstrated that proliferating cells predominantly use transaminases to generate αKG, whereas in quiescent cells GDH dominates [112]. The αKG enters the TCA cycle to generate energy. Additionally, in MM cells αKG is converted into the oncometabolite 2-hydroxyglutamate (2-HG). Elevated 2-HG levels were associated with disease progression in MM, whereas this was not the case for the glutamate and αKG levels. Interestingly, MYC expression correlated with 2-HG levels, which may be related to MYC-driven glutamine anaplerosis [113]. Typically, 2-HG production is catalyzed by mutant isocitrate dehydrogenase 1/2 (IDH1/2), which is a hallmark feature of gliomas and secondary glioblastomas [114], but is very rare in MM [115]. The exact mechanism of 2-HG production in MM remains unknown. It was demonstrated that 2-HG activates the mammalian target of rapamycin (mTOR) by decreasing the protein stability of the DEP domain-containing mTOR-interacting protein (DEPTOR), which is a negative regulator of mTOR [116]. Activation of mTOR is associated with AKT kinase activity and downstream FOXO inactivation. Based on these observations we speculate that elevated 2-HG in MM may be part of a positive feedback mechanism. Oncogenic MYC drives the expression of 2-HG, which results in mTOR/AKT-mediated downmodulation of FOXO that enables further MYC deregulation [63]. In addition, in gliomas it was shown that 2-HG is an inhibitor of αKG-dependent dioxygenases, including histone demethylases, and was associated with histone and DNA methylation changes [117]. It is conceivable that 2-HG may have a similar impact on the epigenome in MM cells and thereby contribute to gene deregulation.

In several cancer cell lines it was demonstrated that glutamine promotes proliferation independently of glutaminolysis, as the effects of glutamine depletion could not be rescued by supplementation with the glutaminolysis intermediates glutamate or 2-oxoglutarate. Glutamine was shown to activate signal transducer and activator of transcription 3 (STAT3), which was not affected by inhibition of glutaminolysis, suggesting that glutamine may be directly involved in transmembrane receptor engagement [118]. The Janus activated kinase (JAK)/STAT3 signaling pathway plays a prominent role in MM, which is activated by interleukin-6 and by loss of Src homology phosphatase -1 (SHP-1) [119,120,121]. In a murine retroviral transduction model, it was shown that expression of a constitutive active IL-6 signal transduction chain induced an MM-like disease that closely mimicked human MM pathology. Importantly, these tumors harbored *Myc* aberrations indicating that STAT3 activation and MYC collaborate to induce MM [122]. Perhaps in MM, glutamine does not only fulfil metabolic demands that are imposed and regulated by MYC, but may also act in a direct fashion by deregulating oncogenic signaling.

### 2.3. MYC and Lipid Metabolism in MM

Proliferation poses an increased demand on lipid synthesis for biogenesis, and fatty acids play a crucial role in the generation of signaling intermediates and fulfil an important function as catabolic reservoir in normal and in cancer cells. MYC is involved in several aspects of fatty acid synthesis and oxidation as shown by metabolic tracing experiments in *Myc^−/−^* and *Myc^+/+^* rat fibroblasts [123]. Furthermore, metabolomic studies in MYC-driven lymphomas identified specific alterations in lipid composition compared to normal tissue [124]. Interestingly, in a comparative study on the fatty acid composition in plasma from MM patients and healthy controls it was demonstrated that saturated and n-6 polyunsaturated fatty acids were increased in MM patients, likely reflecting elevated endogenous production by MM cells. Through its effects on the TCA cycle, MYC promotes the production of citrate and drives the expression of ATP citrate lyase (ACLY) [125], which catalyzes the formation of acetyl-CoA, an important step in fatty acid synthesis. Moreover, MYC also regulates the expression of acetyl-CoA carboxylase 1 (*ACC1*), fatty acid synthetase (*FASN*) and stearoyl-CoA desaturase 1 (*SCD1*), which are involved in fatty acid synthesis [123,126]. Of interest, an RNAi-based loss of function screen showed that ACC1 is required for MM cell survival, and ChIP-seq experiments showed that MYC was bound to the *ACC1* promoter in MM cells [127]. The expression of *SCD1* was found to be upregulated in MM patients and correlated with a proliferation gene signature dictated by MYC, underscoring the functional association between fatty acid synthesis and MYC in MM (Figure 2, Table A1). Small molecule inhibitors for SCD1 decreased proliferation but had a moderate effect on MM cell survival [128]. In addition, inhibition of fatty acid oxidation using the carnitine palmitoyl-transferase 1a (CPT1a) inhibitor etomoxir induced cell death in MM cell lines, indicating that MM cells could also use fatty acids to fuel mitochondrial respiration [129].

The central role of MYC in fatty acid synthesis requires the nutrient-sensing transcription factor MondoA, and knockdown of MondoA induced cell death in several cancer cell lines, showing a synthetic lethal interaction with the expression of MYC [130]. These results indicate that MondoA may also be part of the cellular response to tolerate MYC deregulation. Whether MondoA is involved in the increased fatty acid synthesis in MM remains to be established. MYC also induces the expression of sterol-response element-binding protein 1 (SREBP1), which participates in the regulation of genes involved in fatty acid synthesis [126]. SREBP1 is activated by AKT signaling, which is constitutively active in MM. Downstream of AKT signaling, SREBP1 coordinates the metabolic flux from glycolysis to fatty acid synthesis and is critically involved in cell growth [131]. The regulation of SREBP1 is another example of how both AKT and MYC orchestrate the metabolic demands imposed by increased (oncogenic) proliferation and growth, emphasizing the interdependency of AKT and MYC for malignant transformation and maintenance.

In addition to fatty acid regulation, MYC plays a pivotal role in the mevalonate pathway that is responsible for the production of cholesterol and protein prenylation, including geranylgeranylation and farnesylation. MYC increases the expression of the rate-limiting hydroxymethylglutaryl coenzyme A reductase (HMGCR), which is involved in several cancer types [132]. Similar as for fatty acid synthesis, MYC acts in concert with SREBP1 to drive *HMGCR* expression [126]. HMGCR is the target of statins that act to lower cholesterol levels, and statins were shown to induce cell death in several different cancer types, including MM [133,134,135]. It was recently reported that the mevalonate pathway is of crucial importance for survival of the t(4;14)-positive subset of MM, which is characterized by FGFR3 and MMSET expression. Statins specifically induced apoptosis of t(4;14)-positive MM cells; however, the effect appeared to be independent of FGFR3 or MMSET expression, but was related to the activation of an integrated stress response resulting from loss of protein prenylation. MYC may be involved in this phenotype, since it was shown that MMSET stimulated MM cell growth by repressing mIR-126*, which targets MYC [136].

### 2.4. MYC and Nucleotide Metabolism in MM

To sustain increased proliferation, cancer cells are characterized by an increased demand on nucleotide synthesis. In addition here, MYC was shown to be a central regulator that drives the expression of several genes involved in pyrimidine and purine synthesis [137,138] (Figure 2). Nucleotide synthesis requires glycine and aspartate, which may be derived from glucose and glutamine. Glycine is generated from glucose by phosphoglycerate dehydrogenase (PHGDH) and phosphoserine aminotransferase 1 (PSAT1), and the expression of both enzymes is upregulated through MYC [95]. Interestingly, inhibition of PHGDH was recently shown to reduce MM cell growth and was involved in resistance to the proteasome inhibitor bortezomib in MM [139]. The carbamoyl-phosphate synthetase 2, aspartate transcarbamylase, dihydroorotase (CAD) enzyme is involved in initial steps of pyrimidine biosynthesis and was found to be a direct target of MYC [140]. In addition, MYC regulates dihydroorotate dehydrogenase (DHODH), which catalyzes the subsequent step in pyrimidine synthesis. Furthermore, oxidation of dihydroorotate by DHODH provides electrons for the mitochondrial electron transport chain, coupling nucleotide synthesis to ATP production. DHODH is expressed in MM cells and is essential for cell growth, as it was demonstrated that the specific DHODH inhibitor A771726 induced a G1 cell cycle arrest in MM cell lines. Interestingly, A771726 also inhibited AKT signaling [141], which could potentially interfere with MYC expression and activity in MM cells by activating FOXO [65]. In agreement, DHODH inhibitors and DHODH knockdown downregulated MYC expression in MM cells [142].

Purine synthesis critically hinges on the activity of the phosphoribosylformylglycinamide synthetase (PFAS) enzyme. Induced MYC expression in the engineered P493-6 cell line strongly activated the expression of PFAS [137]. Furthermore, ChIP-seq experiments confirmed direct MYC binding to this gene [143]. It was recently shown that de novo purine synthesis is stimulated by the extracellular signal-related kinase (ERK), which phosphorylates PFAS and is required for tumor growth [144]. ERK is constitutively activated in MM and stabilizes MYC protein by direct phosphorylation and by controlling the PI3K pathway [67]. We speculate that by these means ERK and MYC are part of a positive feedback mechanism that drives purine synthesis in MM cells. As described above, MYC is also important for the generation of ribose-5-phosphate, which is the scaffold for purine synthesis.

### 2.5. MYC and Mitochondrial Biogenesis in MM

Glycolysis, glutaminolysis and fatty acid oxidation primarily drive the TCA cycle for anaplerosis and to generate ATP, which takes place in the mitochondria, and are critically controlled by MYC in cancer cells. Mitochondrial mass is subject to homeostatic control, and during cell growth mitochondrial biogenesis is increased. In resting cells, mitochondrial biogenesis is coordinated by peroxisome proliferator-activated receptor gamma coactivator 1-alpha (PGC1α), whereas this is controlled by MYC in proliferating cells [145,146,147]. Furthermore, mitochondrial activity is associated with altered rates of mitochondrial fusion and fission, which were also found to be regulated by MYC. It was shown that re-expression of Myc in *Myc^−/−^* fibroblasts resulted in the increase in mitochondrial mass by favoring mitochondrial fusion over fission [148]. In MM cells, the expression of several genes involved in mitochondrial biogenesis were found to be upregulated compared to normal plasma cells [149]. Of interest, a recent report described an alternate mechanism by which MM cells may fulfil metabolic demands; in co-culture studies it was shown that mitochondria are transferred from bone marrow stromal cells to MM cells, which involved the formation of CD38-dependent nanotubes [150]. It was suggested that the transferred mitochondria might stimulate the function of the MM cell mitochondria, perhaps through mitochondrial fusion regulated by MYC [151]. In addition, the mitochondrial fission factor (MFF) was identified as a transcriptional target of MYC, and was found to be overexpressed in cancer cells. MFF was shown to associate with the voltage-dependent anion channel-1 (VDAC1) and to be of crucial importance for mitochondrial fitness. MMF silencing resulted in the loss of mitochondrial membrane potential and apoptosis [152].

The broad scale effects of MYC overexpression on metabolic genes in MM and the fact that knockdown of these genes, or inhibition of these enzymes, are generally lethal suggests that MYC overexpression results in a metabolic Achilles’ heel that could be exploited therapeutically.

## 3. Cyclin D1 Regulation in MM

Cyclin D1 (*CCND1*) was originally identified as an oncogene in parathyroid tumors and is upregulated in lymphoid malignancies, including MM. Cyclin D1 is a member of the D-type cyclin family, with other members being cyclin D2 and cyclin D3 [153], which are well known for their canonical functions in regulating the G1 to S phase transition of the cell cycle. D-type cyclins form a complex with either cyclin-dependent kinase 4 (CDK4) or CDK6 and phosphorylate the retinoblastoma protein (RB). This leads to the inactivation of RB, which then releases E2 factor (E2F) transcription factors from its repressive function, promoting cell cycle progression. As such, cyclin D1 expression is tightly regulated during the cell cycle in somatic cells, and uncontrolled expression of cyclin D1 results in bypassing of the cell cycle checkpoints and can eventually lead to neoplastic growth [154]. RAS- and JAK/STAT-mediated signaling pathways both increase *CCND1* transcription during the G1 phase under the influence of growth factors, hormones and cytokines [155]. Phosphorylation of cyclin D1 on residue threonine 286 triggers nuclear exclusion, ubiquitination and proteasomal degradation. Glycogen synthase kinase 3b (GSK3B) and p38 mitogen-activated kinase (MAPK) in combination with ERK are known to phosphorylate cyclin D1 on this residue [156,157]. Cyclin D1 degradation is required before the cell can enter S phase as it interacts with proliferating cell nuclear antigen (PCNA) and in this way inhibits DNA replication [158]. GSK3b is negatively regulated downstream of RAS through AKT and ERK, and MAPK is directly downstream of RAS signaling. As such, RAS activation indirectly inhibits cyclin D1 proteolysis [159].

Next to its canonical role in the cell cycle by interacting with CDK4/6, there is evidence that cyclin D1 influences transcription of other genes. Cyclin D1 is known to bind to more than 30 different transcriptional regulators, amongst which are nuclear receptors, chromatin modifying proteins and transcription factors [160]. One interesting example is STAT3, which induces cyclin D1 expression, although cyclin D1 in turn represses STAT3 function [161]. A genetic-proteomic screen in mice revealed that cyclin D1 was found on more than 900 promoter regions and physically interacted with transcription factors that target these regions [162]. As such, cyclin D1 can influence transcriptional regulation, the DNA damage response pathway, tumor migration and several metabolic pathways [163].

Cyclin D1 expression is frequently elevated in cancer, usually through amplification of the *CCND1* locus, which is associated with a poor prognosis in solid tumors [164]. In B-cell lymphomas, cyclin D1 overexpression is often due to the t(11;14)(q13;q32) translocation, juxtaposing the *IGH* locus to the *CCND1* locus. This translocation is found in nearly 100% of mantle cell lymphomas (MCL), but is also one of the most recurrent translocations in MM, where it is found in 15–25% of all cases. In MCL, this translocation arises during VDJ recombination in developing B cells, whereas illegitimate CSR is responsible for this translocation in MM [165]. Regardless of the t(11;14) translocation, nearly 50% of MM cases show cyclin D1 expression, whereas it is not expressed in normal plasma cells. In the remaining 50% of MM cases cyclin D2 or D3 are often overexpressed, usually as a consequence of FGFR3/MMSET and/or MAF dysregulation, suggesting that D-type cyclin overexpression is one of the hallmarks of MM [166,167]. Of interest, MYC was shown to inhibit cyclin D1 through repression of its promotor [168,169]. The dysregulated expression of D-type cyclins in MM appears to be an early event in MM pathogenesis and seems to drive proliferation, as is observed in many other cancers [167,170]. However, there is some debate whether cyclin D1 overexpression impacts prognosis in MM, whereas cyclin D2 overexpression was found to be associated with a worse outcome [171,172,173,174]. There is some evidence that cyclin D1 expression affects cellular metabolism. For instance, it was recently shown that cyclin D1 can augment AKT serine 473 phosphorylation, thereby promoting AKT activation, which is critically involved in the regulation of several metabolic pathways [175]. Interestingly, AKT negatively regulates GSK3β kinase and FOXO transcription factors through phosphorylation, which both inhibit Cyclin D1 expression, essentially creating a feed forward loop [176].

### 3.1. Cyclin D1 and Glucose Metabolism in MM

Cyclin D1 has been implicated in the regulation of glucose metabolism (Figure 2, Table A1). For instance, in an in vivo breast cancer mouse model, it was demonstrated that *CCND1* silencing enhanced glycolysis and mitochondrial activity, whereas *CCND1* overexpression in the mammary glands inhibited not only glycolysis, but also fatty acid synthesis and the expression of several key mitochondrial genes [177]. In B cells, the ectopic expression of cyclin D1 was shown to decrease OXPHOS through a CDK4/6-independent mechanism where cyclin D1 competes with HK2 for binding to the VDAC1, which regulates OXPHOS [178] (Figure 2). This may indicate that cyclin D1 helps to facilitate the shift from OXPHOS to glycolysis observed in MM. This notion is supported by Caillot et al., who suggest that HK2, VDAC1 and cyclin D1 colocalize in the cytosol of the LP-1 MM cell line. In their experiments, overexpression of cyclin D1 in LP-1 cells led to a shift from OXPHOS to glycolysis via increased expression of HK2 [45]. The augmented activation of AKT by cyclin D1 may also drive glycolysis, as was observed in other cancer types [179]. Interestingly, the glycolysis enzyme PKM2 was shown to drive cyclin D1 expression by an epigenetic mechanism involving the release of histone deacetylase 3 (HDAC3) from the *CCND1* promoter [180]. It is conceivable that cyclin D1 and PKM2 constitute a feedforward loop that contributes to the metabolic shift towards glycolysis. Whether such a feedforward loop is operable in cyclin D1-expressing MM cells remains to be established.

Little is known of the effects of cyclin D2 on glucose metabolism, although a clear effect of cyclin D3 has been described. In T-cell acute leukemia (T-ALL) cells, inhibition of the cyclin D3-CDK6 complex resulted in inhibition of phosphofructokinase 1 (PFK1) and PKM2, which redirected glucose metabolites into the PPP and the serine biogenesis pathway [181]. Both these pathways were implicated in resistance to bortezomib treatment in MM patients [167,182]. However, whether cyclin D3 expression in MM is associated with bortezomib sensitivity is unknown.

### 3.2. Cyclin D1 and OXPHOS in MM

Cyclin D1 has also been implicated in the regulation of OXPHOS. In hepatocytes it was shown that insulin activated the cyclin D1/CDK4 complex to phosphorylate the general control non-repressed 5 protein (GCN5), which is an acetyltransferase that targets PGC1α and inhibits its function, resulting in the repression of gluconeogenesis and OXPHOS [183] (Figure 2, Table A1). The cyclin D1-dependent repression of PGC1α was abrogated in the presence of a CDK4 inhibitor, or upon expression of mutant cyclin D1 mutant that could no longer activate CDK4 [184]. PGC1α has been shown to be upregulated in MM cell lines under conditions of high glucose and/or treatment with dexamethasone [185]. Furthermore, PGC1α knockdown in the RPMI-8226 MM cell line resulted in increased expression of the GLUT4 glucose transporter (*SLC2A4*), which may drive glycolysis and OXPHOS due to increased glucose uptake [186]. *PGC1A* expression is increased in MM compared to normal plasma cells and high expression of *PGC1A* in MM was associated with a modest but significant survival disadvantage [177]. These observations could be explained by the effect that cyclin D1 has on PGC1α, possibly by enhancing CDK4-mediated inhibition of PGC1α. Interestingly, *Ccnd1^−/−^* mice showed a two- to three-fold increase in mitochondrial size and activity in hepatocytes and mammary gland adipocytes. Mitochondrial activity was also increased in hepatocytes, mouse embryonic fibroblasts (MEFs) and macrophages from the *Ccnd1^−/−^* mice. It was demonstrated that cyclin D1 inhibited the expression and the activity of the nuclear respiratory factor 1 (NRF1), which indirectly controls mitochondrial DNA replication [187]. Treatment of MM cells with proteasome inhibitors resulted in NRF1 upregulation, which was shown to be involved in resistance to proteasome inhibitors by ramping up proteasome subunit gene expression and potentially mitochondrial biogenesis [188]. Whether this has consequences for OXPHOS in MM cells remains to be established. However, in agreement, it was found that the expression of mitochondrial biogenesis and OXPHOS signature genes were increased in MM cells compared to normal plasma cells. Furthermore, relapsed and drug-resistant MM patients had higher expression of mitochondrial biogenesis signature genes than newly diagnosed patients [149].

### 3.3. Cyclin D1 and Lipid Metabolism in MM

Cyclin D1 has been described to influence fatty acid metabolism. Knockdown and overexpression experiments in hepatocytes and breast cancer cells demonstrated that cyclin D1 represses fatty acid oxidation by inhibiting peroxisome proliferator-activated receptor alpha (PPARα) activity, a nuclear receptor that induces fatty acid oxidation. By use of a cyclin D1 mutant that is unable to activate CDK4, it was shown that CDK4 was not involved in the regulation of fatty acid oxidation [189]. In addition, PPARγ was also reported to be negatively regulated by cyclin D1 [190,191]. PPARγ is mainly expressed in adipose tissue where it stimulates lipid uptake and storage upon activation through binding polyunsaturated fatty acids [192]. Interestingly, PPARγ is also expressed in MM and overexpression of PPARγ, or exposure of MM cells to its ligands, provoked apoptosis in MM cell lines [193,194]. The mechanism of this apoptotic effect remains poorly understood and whether this is particularly toxic for cyclin D1-expressing MM, or involves lipid metabolism, is unknown.

In addition to fatty acid oxidation, cyclin D1 also appears to regulate de novo fatty acid synthesis. In hepatocytes it was shown that cyclin D1 inhibited the expression of several key genes involved in fatty acid synthesis such as *FASN*, pyruvate kinase L/R (*PKLR*) and thyroid hormone responsive (*THRSP*), by inhibiting promoter recruitment of the carbohydrate response element binding protein (chREBP) and hepatocyte nuclear factor 4a (HNF4A), which both critically regulate lipid metabolism [195]. Of interest, activation of *HNF4A* expression was shown to be associated with proteasome inhibitor resistance in MM cell lines, but whether this impacted on lipid metabolism is unknown [196]. Importantly, serum metabolomics indicated a significant change in lipid composition in MM patients compared with healthy donors [197,198], but the potential relation with cyclin D1 expression was not explored. From these results it is clear that next to its canonical role in the regulation of cell cycle progression cyclin D1 also has a role in the regulation of cellular metabolism, similar to MYC, albeit to a lesser extent, and with some striking differences.

## 4. FGFR3 Deregulation in MM

Around 10–15% of MM patients harbor the t(4;14)(p16;q32) translocation that results in overexpression of the receptor tyrosine kinase FGFR3 and the histone methyltransferase MMSET [1,2]. Although overexpression of MMSET is universal in t(4;14) [199], approximately 30% of t(4;14) patients is FGFR3 negative [4,5]. Interestingly, the t(4;14) translocation is associated with poor prognosis in MM regardless of FGFR3 expression [6,7]. FGFR3 is a receptor tyrosine kinase that belongs to the FGFR-family of growth factor receptors. Twenty-two members of the FGF family have been identified in humans and a large part of the functional specificity of FGFR activity is determined by binding of distinct FGFs. In accordance, knockout of different *Fgf* family members has diverse and distinct consequences in mice and some *Fgf* knockouts are embryonic lethal (reviewed in [200]). FGFs can function in an autocrine and/or paracrine fashion, but can also mediate FGFR-activation over longer distances [200]. In addition to FGFR3 overexpression in t(4;14) MM, mutations in the *FGFR3* gene have also been reported in several cancers, including MM [201,202,203,204]. The most common mutations found in *FGFR3* occur in the regions coding for the extracellular and transmembrane parts of the protein, resulting in increased receptor dimerization and ligand-independent signaling [205]. Regardless of mode of activation, FGFR engagement results in activation of MAPK-signaling through FGFR-substrate 2 (FRS2) mediated recruitment of son of sevenless (SOS) and growth factor receptor-bound 2 (GRB2). Recruitment of GRB2-associated binding protein 1 (GAB1) to the FRS2 complex results in activation of the PI3K/AKT pathway. Furthermore, binding of phospholipase Cγ (PLCγ) to the carboxy-terminal tail of autophosphorylated FGFR stimulates the release of intracellular calcium which results in activation of the protein kinase C (PKC) family of proteins. Moreover, FGFRs can activate the JAK-STAT pathway, further diversifying FGFR signaling [14,15].

### 4.1. FGF/FGFR and Glucose Metabolism in MM

Although MMSET and FGFR3 overexpression are not likely to be directly linked in t(4;14) MM, activity of either protein can influence MM cell metabolism. Interestingly, FGFR-amplified cells were dependent on glucose instead of glutamine for growth, a trait shared with epithelial growth factor receptor (EGFR)-amplified cells. Wildtype cells lacking these gene alterations showed variable dependency on glucose or glutamine for sustaining growth [206]. Similar to most RTKs, FGFR3 can activate the MAPK-ERK [207] signaling pathway and the PI3K-AKT pathway [208,209], sustained signaling through which can influence metabolic flux by the canonical metabolic master regulators MYC and mTORC, respectively (Figure 3). However, the FGF/FGFR axis exerts metabolic control beyond the regular signaling functions of a receptor tyrosine kinase. Indeed, FGF signaling is a critical regulator of vascular development through regulation of HK2 [210]. In this context, loss of FGFR signaling resulted in hampered HK2 levels, decreased glycolysis and impaired proliferation and migration of endothelial cells. Additional evidence that FGFR may be involved rewiring glucose metabolism was provided by overexpression of *FGFR* in the Ba/F3 mouse pro-B cell line. Functional metabolomics showed that FGFR overexpression enhanced glycolysis and at the same time upregulated consumption of extracellular lactate to fuel OXPHOS (Figure 2, Table A1). Interestingly, glucose and lactate had an equal contribution in fueling OXPHOS in *FGFR*-overexpressing cells. Moreover, several glycolysis genes such as *LDHA*, *HK2*, *GLUT1*, and ATP-dependent 6-phosphofructokinase (*PFKL*) were shown to be upregulated in FGFR-overexpressing cells, resulting in an increased dependence on glycolysis. Mechanistically, these metabolic changes involved HIF1α and MYC, as both were found to be upregulated in FGFR-overexpressing cells, and conversely, downregulated when FGFR was inhibited [206]. Based on these results we speculate that MYC and FGFR3 aberrations have overlapping functions in MM, and may collaborate to skew the metabolic features of MM cells. In agreement, it was shown that *Fgfr3* and *Myc* collaborated to drive mouse lymphomagenesis in a transgenic model [211]. The potential functional connection between FGFR3 and MYC in MM is also apparent from a recent study where it was shown that autocrine FGF/FGFR signaling was shown to protect MM cells from oxidative-stress induced apoptosis [212]. In this paper, the authors show that blocking FGF/FGFR3 signaling through trapping of FGF ligands or by inhibiting the FGFRs leads to proteasomal degradation of MYC through activation of GSK3, which is negatively regulated by AKT. The authors confirmed the role of MYC in this process by overexpressing a non-degradable MYC mutant, which prevented FGF-trapping induced production of mitochondrial reactive oxygen species (mtROS) and apoptosis.

Based on these observations it is likely that FGFR3 signaling in MM cells would also result in increased glutaminolysis by upregulation of MYC expression and activity, although this was not apparent in FGFR-overexpressing Ba/F3 cells [206].

### 4.2. FGFR3 and Lipid Metabolism in MM

FGFR3 was shown to play a specific role in de novo fatty acid and sterol biosynthesis and metabolism [213] (Figure 2, Table A1). FGFR3 knockdown and gene expression profiling in bladder cancer cells revealed a gene signature linking FGFR3 signaling to fatty acid synthesis. Signaling through FGFR3 led to the cleavage and activation of SREBP1 in a PI3K/AKT/mTORC1 dependent manner. SREBP1 regulates the expression of several lipogenic enzymes, including SCD1. Knockdown of SCD1 reduced cell cycle progression and proliferation and induced apoptosis in a bladder cancer xenograft model [213]. Thus, FGFR3 steers the biosynthesis of monounsaturated fatty acids in bladder cancer, suggesting that FGFR3 may potentially also regulate lipid homeostasis in MM. Recent insights into lipid metabolism in MM indicate an important role for altered lipid homeostasis, as increased acid sphingomyelinase (ASM, which converts sphingomyelin to ceramide) expression contributes to drug resistance to melphalan and bortezomib, a drug-resistance phenotype that could be transferred to chemosensitive cells through exosomes [214]. Of note, ASM activity in MM cells can be stimulated by epigallocatechin-3-O-gallate-induced production of cyclic guanosine monophosphate (cGMP), leading to activation of PLCγ and subsequent activation of protein kinase Cδ (PKCδ), suggesting FGF/FGFR signaling can induce ASM activity through PLCγ [215]. Indeed, FGFR3 mutants with reduced capacity for PLCγ activation strongly attenuated, but not abolished, transformative capacity in Ba/F3 cells and in a bone marrow murine leukemia transplant model, underling the importance of PLCγ for oncogenesis downstream of FGFR3 [216]. Interestingly, PLCγ can activate AMPK in a Ca^2+^-dependent manner that involves mTOR, providing an additional layer of metabolic control to FGF/FGFR signaling [217,218]. Importantly, FGF2 and FGF8, two principal ligands for FGFR3, potently enhanced PLCγ phosphorylation and downstream signaling in MM cells, showing that FGFR3-mediated PLCγ activation is operable in MM cells [219]. Whether AMPK may be activated upon FGFR3 engagement on MM cells remains to be established, but it must be noted that AMPK activation inhibited the growth of MM cell lines by negatively regulating AKT [220], making it unlikely that AMPK signaling is involved in the oncogenic effects of FGFR3 in MM cells.

Next to fatty acid synthesis, the mevalonate pathway was also implicated in t(4;14) MM. It was demonstrated that inhibition of the rate-limiting enzyme HMGCR by statin treatment is preferentially toxic to t(4;14) MM cells [221]. Inhibition of HMGCR through fluvastatin activated the integrated stress response (ISR), a stress response induced by ER stress and nutrient deprivation, amongst other things. Activation of this response resulted in phosphorylation of the eukaryotic translation initiation factor 2A (eIF2a) and subsequent attenuation of global mRNA translation. Simultaneously, phosphorylation of eIF2a allows for the selective translation of proteins involved in cell survival and recovery, such at ATF4. Experiments assessing glucose regulated protein 78 (GRP78) and XBP1 splicing (integral parts of the UPR) showed that fluvastatin treatment induced ATF4 target gene expression in an ER-stress independent manner, as treatment with fluvastatin treatment had no effect on GRP78 and XBP1 levels. Statin-induced apoptosis could be prevented by addition of the mevalonate pathway product geranylgeranyl pyrophosphate (GGPP) but not by supplementation with farnesyl pyrophosphate (FPP). Treatment of t(4;14) MM cells with geranylgeranylation inhibitor induced ISR-related gene expression, but this response was absent in non-t(4;14) MM cells. The authors speculate that t(4;14) MM cells require the mevalonate pathway activity, in part, for protein prenylation.

## 5. MMSET Deregulation in MM

MMSET, also known as nuclear receptor-binding SET domain 2 (NSD2) and Wolf-Hirschorn Syndrome Candidate 1 (WHSC1), regulates chromatin integrity and gene expression through methylation of histone lysine residues [222] and the its primary chromatin-regulating catalytic activity is dimethylation of histone H3 at lysine 36 (H3K36me2), resulting in a more open chromatin state [223,224]. Another consequence of increased MMSET expression is a global reduction of trimethylation of lysine 27 on Histone H3 (H3K27me3) [225]. Accordingly, the t(4;14) translocation in MM induces increased lysine 36 and decreased lysine 27 methylation across the genome, resulting in a more accessible chromatin state and changes in expression of proteins involved in integrin signaling, p53 signaling and cell cycle regulation [225,226,227,228]. Interestingly, certain genomic loci actually show an increase in H3K27 methylation and enhanced recruitment of enhancer of zeste homolog 2 (EZH2), leading to transcriptional repression of genes located on these loci. MMSET overexpressing cells showed high sensitivity to compounds inhibiting EZH2, suggesting that repression of these genes is important for MMSET overexpressing tumors [229]. Hemizygous loss of *Mmset* caused a significant decrease in B- and T-lymphocyte numbers in aging mice [230]. Competitive fetal liver transplant experiments showed that all types of *Mmset^−/−^* blood cells were outcompeted compared to wildtype cells, indicating a developmental disadvantage in *Mmset^−/−^* lineage cells. Serial transplants of *Mmset^−/−^* cells showed reduced repopulation potential after tertiary transplants as compared to wildtype cells. Furthermore, secondary transplant recipients had very few T cells and no B cells, suggesting severe B-cell developmental deficiency upon *Mmset* deletion. Moreover, immunoglobulin class switching efficiency was reduced in MMSET null cells due to a proliferative defect and an increase in apoptosis. Further analysis showed cell-cycle progression defects, defects in ribosome synthesis, splicing defects and decreased DNA-damage repair capacity [230]. Moreover, MMSET was reported to increase AID-mediated DNA breaks in donor switch regions during CSR, underlining the importance of functional MMSET in B-cell development as a critical mediator of antibody diversification and immune function [231]. MMSET was found to be involved in DNA repair through its histone methylation activity that stimulates non-homologous end-joining and by promoting AID-instigated DNA breaks, which are both required for class switching [231,232]. Knockdown of MMSET in U2OS cells led to decreased protein expression of the DNA repair proteins RAD51 and 53BP1, suggesting that MMSET may also play a role in the transcriptional control of DDR proteins. Finally, knocking down MMSET in a mouse MM xenograft model strongly increased melphalan sensitivity. MMSET thus drives increased DDR, thereby increasing resistance to chemotherapeutic drugs [233].

### Metabolic Consequences of MMSET Deregulation in MM

MMSET also modulates a diverse range of metabolic processes, often in a context-dependent manner. Tamoxifen-resistant breast cancer cell lines showed increased MMSET expression and concomitant elevated expression of key glycolytic enzymes HK2, G6PD and TIGAR as a consequence of promoter dimethylation at H3K36 by MMSET [234]. The MMSET-mediated increase in glycolysis concomitantly heightened the activity of the PPP, which is of vital importance for cellular redox balance in B cells [235,236] (Table A1). Another interesting feature of MMSET transcriptional control is prevention of senescence, which is linked to profound metabolic changes. An siRNA based screen of chromatin regulators uncovered that MMSET knockdown induced senescence in primary human fibroblasts and increased mitochondrial mass and OXPHOS [237]. Interestingly, reduced expression of MMSET was observed in oncogene-induced and replicative-senescent cells, coupled with diminished H3K36me3 levels. Interestingly, MMSET expression levels were correlated with cell-cycle related genes and expression of MMSET was upregulated by serum stimulation, due to direct phosphorylation of MMSET by AKT [237]. AKT stabilized MMSET protein expression and provoked the MMSET-dependent expression of RICTOR, an important component of mTORC2, thereby further enhancing AKT-activity [238].

A truncated isoform of MMSET (MMSET I) was shown to possess a specific metabolic function in t(4;14). MM [239]. MMSET I drives expression of glyoxalase 1 (GLO1), a crucial enzyme in the detoxification of the aldehydes that are normal byproducts of cellular metabolism [240]. ChIP analysis showed that *GLO1* is a direct transcriptional MMSET I target. Interestingly, GLO1 knockdown induced apoptosis and reduced colony formation potential of t(4;14) myeloma cells, mainly by reducing the levels of the antiapoptotic proteins MCL1 and BCL-2. Furthermore, MMSET I knockdown reduced glycolysis, which was partially rescued by ectopic overexpression of GLO1 [239]. Taken together, these results indicate that MMSET activity drives glycolysis and PPP-activity, while inhibiting OXPHOS.

## 6. MAF Deregulation in MM

The t(14;16) and t(14;20) translocations deregulate *c-MAF* and *MAFB* and are relatively rare events in MM [241,242,243]. In t(14;16) MM, this results in overexpression of the transcription factor MAF and in overexpression of transcription factor MAF homolog B (MAFB) in t(14;20) MM. An even smaller group of MM cases harbor a translocation juxtaposing the *MAFA* gene, leading to aberrant expression of MAFA [241,244]. Different MAF translocations confer a similar prognosis in newly diagnosed MM patients, possibly due to functional redundancy of MAF proteins in MM [245]. The MAF transcription factors belong to the AP1 -super-family of basic leucine zipper proteins, a family that also includes the transcription factor families FOS, JUN, CREB and ATF [244]. Transcription activation by large MAF proteins is achieved by recruiting p300/CBP-associated factor (P/CAF) [246], p300 (also known as EP300) [247] or tata-box binding protein (TBP) [248]. Transcriptional activation by the large MAF proteins (MAFA, MAFB, c-MAF and NRL) is negatively regulated by homodimers of the small MAF proteins (MAFF, MAFG and MAFK) through competitive binding [244], indicating that the ratio of small/large MAF proteins is an important determinator of MAF-mediated gene regulation [249]. Under physiological circumstances and during development, expression of MAF proteins is restricted and tightly regulated [244,250,251]. Stringent spatio-temporal regulation of expression is important, since MAF proteins play important roles in tissue specification and differentiation [252,253]. Several posttranslational modifications also play a role in controlling MAF protein activity, such as phosphorylation by GSK3 [246,254], through FGFR/ERK signaling [255] and by p38 MAP kinase [256]. Phosphorylation by GSK3 promotes ubiquitination and degradation of MAF but also stimulates recruitment of co-activator P/CAF, enhancing transactivation activity of MAF and protecting the protein from degradation [246,254]. Stability of MAF proteins in MM is further regulated by the deubiquitinases ubiquitin-specific peptidase 5 (USP5), which prevents degradation of c-MAF and MAFB [257], and USP7, which can deubiquitinate c-MAF, MAFA and MAFB [258].

The large MAF genes are oncogenes and have transformative capabilities in a wide range of tissue backgrounds, such as in primary fibroblasts [259,260]. Moreover, c-MAF is overexpressed in 60% of angioimmunoblastic T-cell lymphomas in humans [261] and expression of c-MAF in the T-cell compartment in transgenic mice results in development of T-cell lymphomas [262]. Transgenic mice with *Maf* expression targeted to B cells develop B-cell lymphoma with features resembling MM but fail to fully recapitulate human-like MM [263]. However, expressing MAFB in hematopoietic stem/progenitor cells caused the development of plasma cell neoplasia that recapitulated human disease [264], a phenotype that was accelerated by loss of *Tp53* [265]. Interestingly, overexpression of c-MAF is not exclusive to MM cases that harbor t(14;16), and the percentage of MM cases that overexpress *MAF* is around 50%, mainly because of high expression in the t(4;14) and the t(11;14) MM cases [266,267,268], although the transcriptional consequences of MAF expression in non-translocated MM are modest as judged by gene expression-based clustering [269]. Interestingly, the MMSET and MAF groups show similar gene expression profiles [269]. Overexpression of *MAF* in MM that is not attributable to a translocation can be driven by TNFRSF13B (also known as TACI) and/or XBP1, as high expression of *TNFRSF13B* is associated with high expression of *MAF* and lymphoid-specific XBP1 transgenic mice show deregulated *c-MAF* and *MAFB* expression [270,271]. Expression of MAF proteins can also be regulated by MEK/ERK activity, as inhibition of MEK induced downregulation of *MAF* mRNA in MMSET/MAF translocated MM [272]. Of note, t(14;16) and t(14;20) were strongly associated with an apolipoprotein B mRNA editing enzyme (APOBEC) mutational signature and knockdown of c-MAF or MAFB resulted in downregulation of apolipoprotein B mRNA editing enzyme (APOBEC) proteins, suggesting a reciprocal relation between MAF and APOBEC that drives ongoing mutagenesis in these MM subsets [273,274].

Another important oncogenic role of MAF proteins in MM is attenuating proteasome inhibitor toxicity. Patients with t(14;16) MM do not benefit from the addition of bortezomib to standard therapy regimens when compared to other MM subsets [275]. Indeed, high expression of *MAF* is associated with innate resistance to proteasome inhibitor treatment [276,277,278] and silencing *MAF* sensitized MM cells to proteasome inhibitors, while proteasome inhibition abrogated GSK3β-mediated degradation of c-MAF, effectively stabilizing MAF activity [276,278]. In accordance, overexpression of c-MAF induced resistance to proteasome inhibitor treatment [276].

### 6.1. MAF Target Genes, Associated Mutations, and Metabolic Effects of MAF

MM cases that carry *MAF* translocations share a distinct gene expression signature that is similar between different MAF proteins [115,279]. In a cohort of 1273 newly diagnosed patients, the t(14;16) translocation was associated with mutations *BRAF*, *TRAF2* and *DIS3* (mutations that affect MAPK-signaling, NFKB signaling and exosome formation), deletion of 13q, gain of 1q and the aforementioned APOBEC signature [115]. Genes strongly associated with *MAF* overexpression in MM include *CCND2*, integrin-β7 (*ITGB7*), C-C chemokine receptor-1 (*CCR1*) [266,280], AMPK-related protein kinase 5 (*ARK5*) [281] and *DEPTOR* [268]. MAF can therefore directly regulate proteins involved in cell cycle regulation (*CCND2*), MM cell homing, migration and adhesion (*ITGB7, CCR1*), insulin growth Factor-1 (IGF1) mediated invasion (ARK5) and PI3K/AKT/mTORC signaling (DEPTOR) (Figure 3). This suggests a potentially multifaceted role for *MAF* family members in metabolic control.

### 6.2. MAF and Glucose Metabolism in MM

MAF proteins play an important role in the regulation of insulin secretion. MAFA, in conjunction with the insulin promoter factor 1 (PDX1) transcription factor, can stimulate production and secretion of insulin in response to glucose by reprogramming islet non-β-cells [282]. High levels of glucose, but not pyruvate, can stimulate *MAFA* expression in pancreatic β-cell lines in a mechanism dependent on the HBP, a pathway linked to glycolysis via fructose-6-phosphate [283]. As such, the HBP links nutrient sensing and energy metabolism to posttranslational protein modifications, indicating that HBP flux and cellular metabolism are tightly associated [284]. Treatment with the chemotherapeutic drugs doxorubicin and camptothecin activated the HBP in an AKT/XBP1 dependent manner [285]. Interestingly, increased activity of the HBP is associated with resistance to proteasome inhibitors through differing mechanisms. Stabilization of NRF1 by O-linked N-acetylglucosamine transferase (OGT), a protein that catalyzes the addition of GlcNAc moieties, elevated the expression of several proteasomal subunit genes, leading to proteasome inhibitor resistance in breast cancer and non-small cell lung cancer cell lines [286]. In another study, bortezomib-resistant U266 MM cells were shown to have increased numbers of mitochondria and elevated levels of the mitochondrial biogenesis markers PGC1α and the NAD-dependent deacetylase sirtuin 1 (SIRT1) [287]. Quantification of glycosylated UDP-containing compounds revealed the increased presence of glycosylated products in the bortezomib-resistant cells, indicating increased HBP activity. Bortezomib treatment did not induce mitochondrial depolarization in resistant cells indicating that retaining mitochondrial function is important for bortezomib resistance, a process possibly mediated by increased protein glycosylation, although critical downstream determinants of this process remain unclear [287]. Based on these findings, coupled with the clinical observation that MAF translocated MM shows innate resistance to proteasome inhibition, we speculate that increased activity of the hexosamine pathway can stabilize MAF proteins, a function that could be important regardless of overexpression through translocation, and that activity of the HBP through glycolytic intermediates is associated with stabilization of NRF1 and increased mitochondrial fitness.

Another aspect of glucose metabolism in relation to MAF proteins is coupled to the nutrient sensing functions of the cell cycle machinery. Glucose was shown to regulate *CCND2* mRNA levels in quiescent and replicating pancreatic β-cells through glycolysis and calcium channel metabolism [288]. Stimulation of glycolysis by activating glucokinase increased Cyclin D2 levels as did calcium-influx. Experiments with *Mafa* knockout mice shows that MAFA controls expression of the calcium channel subunit gamma-4 in β-cells, which is vital for glucose-induced insulin exocytosis [289]. Of note, *Ccnd2* knockout mice show a diabetic phenotype due to dysfunctional β-cell outgrowth [290]. Although evidence in MM is lacking, this raises the possibility that abundant glucose or increased glycolytic flux can activate cyclin D2 in MM cells through activity of MAF proteins, perhaps in conjunction with the HBP.

ARK5 is an important downstream target of MAF proteins in MM as it regulates invasion of cancers cells downstream of AKT by upregulating several matrix metalloproteases [291] and in MM, ARK5 mediates invasion through IGF1 [281]. The MAF target gene *ARK5* was implicated in maintaining mitochondrial function in neuronal cells and could therefore be involved in the regulation of OXPHOS. In agreement, deletion of ARK5 in neuronal cultures decreased both the basal and maximal respiratory rate but did not affect glycolysis, indicating a defect in OXPHOS [292]. Interestingly, ARK5 possesses distinct cytosolic and nuclear functions, and subcellular localization of the protein differs between cell lines and redox state, as oxidative stress can induce cytosolic accumulation of ARK5 [293]. Cytosolic ARK5 maintained ATP levels in hepatocarcinoma cells by increasing mitochondrial respiration and by regulating mitochondrial morphology [294]. Chemical inhibition of ARK5 decreased the maximal respiration and spare respiratory capacity of MCF-7 breast cancer cells and inhibition or knockdown of ARK5 led to an increase of in the mitochondrial membrane potential [294]. ARK5 represents an interesting clinical target in MM, as dual inhibition of ARK5 and CDK4 in MM by the inhibitor ON123300 led to swift induction of cell cycle arrest, followed by apoptosis. Treatment with ON123300 or knockdown of ARK5 using siRNA induced activation of AMPK and the deacetylase SIRT, while downregulating mTOR/AKT signaling. A functional consequence of this was the downregulation of key cell cycle regulators like cyclin D1, cyclin E, CDK4 and MYC. However, translocations in MAF were not predictive for sensitivity to ARK5/CDK4 inhibition [295]. Although metabolic effects of ARK5/CDK4 inhibition were not investigated here, we speculate that this can lead to diminished OXPHOS activity through abrogated activity of ARK5, possibly through altered mitochondrial morphology, and increased activation of PGC1α through diminished CDK4-dependent activition of GCN5 (Table A1).

High level of serum interleukin-10 (IL-10) is a predictor of poor prognosis in MM [296], and IL-10 was shown to induce proliferation and angiogenesis in MM [297]. Interestingly, MAF is critical for the regulation of IL-10 in immune cells [298,299,300,301]. IL-10 abrogates lipopolysaccharide-induced glucose uptake and glycolysis and promotes OXPHOS in macrophages. Importantly, IL-10 was shown to increase OXPHOS in B cells activated with TLR ligands [302]. Based on these observations we speculate that MAF-driven production of IL-10 in MM cells may constitute an auto- and paracrine feedback loop that broadly inhibits protein synthesis through mTOR, downregulates glycolysis and stimulates OXPHOS [303].

### 6.3. MAF and Glutamine Metabolism in MM

MAF-regulated ARK5 expression could also have consequences for glutamine metabolism, since glutamine was increased upon depletion of ARK5. Metabolic flux assays using ^13^C-glucose and ^13^C-glutamine revealed that the flow of glucose into the TCA-cycle was low and that most glucose was instead converted to lactate, indicating a shift towards glycolysis. In this context, depletion of ARK5 had little effect on glycolysis, but rather increased flow from ^13^C-glutamine to αKG, consistent with increased consumption of glutamine. However, ARK5 depletion prevented αKG entry into the TCA-cycle, as indicated by diminished levels of succinate, fumarate and malate. Proteomic analysis of ARK5 depleted cells showed downregulation of multiple subunits of respiratory chain complexes I, III and IV. As a consequence, ARK5 depletion led to decreased oxygen consumption. Taken together, this shows an important role for ARK5 in sustaining glutamine metabolism by maintaining sufficient respiratory capacity [304]. Functionally, inhibition of ARK5 in the context of deregulated MYC leads to global accumulation of RNA polymerase II (RNAPII) at the pause site and the first intron-exon boundary, but does not increase mRNA synthesis, indicating a defect in MYC-regulated gene expression upon loss of ARK5 functionality [305]. This gene regulation defect could explain the eventual failure of ARK5 depleted cells in sustaining TCA-cycle driven metabolism through failure of glutamine utilization. As MYC deregulation is a hallmark of MM, overexpression of MAF proteins and downstream upregulation of ARK5 can ensure MM cell survival through maintenance of glutamine metabolism.

### 6.4. MAF and Lipid Metabolism in MM

The MAF-dependent regulation of IL-10 expression in immune cells was reported to be dependent on the cholesterol biosynthesis pathway. Inhibition of cholesterol biosynthesis through statin treatment or supplementation of 25-hydroxycholesterol induced a significant decrease in IL-10 expression. [306]. Of note, physiological levels of 25-hydroxycholesterol significantly decreased MAF expression in these cells, suggesting that careful balancing of cholesterol synthesis is important for MM cells that require high MAF activity. However, whether cholesterol metabolism converges with MAF deregulation in MM remains to be established.

## 7. Concluding Remarks and Future Perspective

MM is a genetically heterogeneous disease, which is reflected by an equally heterogeneous clinical course and prognosis. The differential behavior and specific therapeutic vulnerabilities of MM cells are in large part dictated by a handful of recurrent chromosomal aberrations that to a sizeable extent demarcate prognostic patient subgroups. Recent progress in the molecular characterization of MM allows the deep interrogation of oncogenic mutations and (epi)genetic changes, which will further forward the elucidation of the etiology, pathobiology, and furthermore, provide new means to stratify patients for personalized therapy. From that point of view, integration of knowledge on the potential metabolic consequences of oncogenic events will provide an even more fine-grained understanding of the altered biology of MM cells in comparison to its normal counterparts, and perhaps uncover novel dependencies that may be amenable to therapeutic targeting. In that respect, an effort towards the comprehensive metabolic profiling of the MM genetic subgroups would be both timely and insightful to obtain a more holistic view on the cellular consequences of oncogenic events in MM. In comparison to most other cancer types, the secretory nature of MM cells stands out. In contrast to normal plasma cells, MM cells are characterized by antibody secretion (in most cases) and increased proliferation, whereas normal plasma cells show very limited proliferative capacity. The chromosomal aberration involving cyclin D1 and MYC are firmly associated with increased proliferation in MM, but strikingly, are also involved in metabolic changes that may not be directly linked to proliferation. It appears that MYC is a dominant factor in altering or amplifying metabolic features of MM plasma cells (Figure 3). Even in MM cells that do not carry MYC aberrations, other chromosomal events may indirectly result in the activation of stabilization of MYC and thereby influence metabolism. As is the case for most others cancers, MM is associated with the increased uptake of glucose and glutamine, which are metabolized by glycolysis and glutaminolysis, respectively, to provide precursors for anaplerotic reactions and to fuel OXPHOS for energetic demands. Glucose and glutamine are also required for antibody glycosylation, which further increases the demands of these nutrients. Both glycolysis and glutaminolysis are critically driven by MYC, but cyclin D1, FGFR3/MMSET and MAF also appear to enhance these metabolic pathways. It is clear that there is a considerable degree of redundancy between these oncogenes with regards to their metabolic effects (Figure 3). The fast rate by which glycolysis can provide both energy and biosynthetic precursors may underlie the increased reliance of MM cells on this pathway. The constitutive activation of the AKT pathway, which is characteristic of MM cells, may be a crucial metabolic driver in this context, as it ramps up most metabolic pathways, while at the same time allowing the full activation of oncogenes such as MYC, thereby further boosting the metabolic activity of MM cells. Additionally, OXPHOS is clearly also increased in MM cells, as it is linked to mitochondrial activity and apoptotic thresholds of MM cells. This is exemplified by the observation that proteasome inhibitors, which can be considered breakthrough drugs for the treatment of MM, provoke changes in mitochondrial integrity. Furthermore, resistance to proteasome inhibitors is associated with metabolic rewiring that increases OXPHOS capacity. As treatment efficacy and resistance are linked to metabolic plasticity of cancer cells, it will be of importance to carefully map the metabolic flexibility in the context of therapy. MM cells display a striking degree of clonal heterogeneity, which may cause metabolic plasticity and could potentially allow MM cells to become resistant to treatment. Characterization of (epi)genetic changes and accompanying metabolic alterations in treatment resistant MM cells will provide further insight into how mutagenic events shape the metabolic features of these cells.

## Figures and Tables

**Figure 1 cancers-13-00396-f001:**
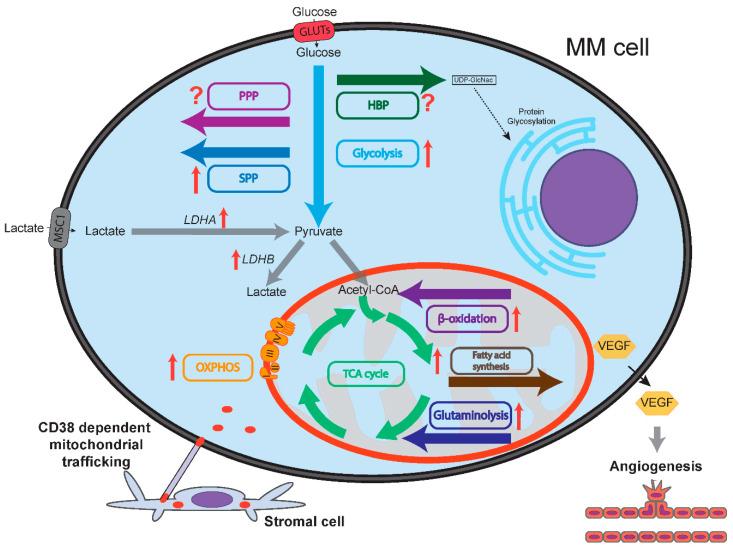
Schematic representation of multiple myeloma (MM) cell metabolism compared to normal plasma cells. Flow of metabolites is shown by colored arrows. Glycolysis, OXPHOS, serine synthesis pathway (SSP), glutaminolysis, fatty acid synthesis and fatty acid oxidation (β-oxidation) are all upregulated in MM cells compared to normal plasma cells, as indicated by red arrows. Lactate dehydrogenase A (*LDHA*) and B (*LDHB*) are upregulated in MM cells. It is unclear whether the pentose phosphate pathway (PPP) and the hexosamine biosynthesis pathway (HBP) are up/downregulated in MM cells compared to normal plasma cells, as designated by question marks. MM cells express vascular endothelial growth factor (VEGF) that increases bone marrow oxygenation by inducing angiogenesis. Increased bone marrow oxygenation may stimulate oxidative phosphorylation (OXPHOS) in MM cells. CD38-dependent nanotubes transport mitochondria from stromal cells to MM cells, mitochondrial fusion further sustains OXPHOS.

**Figure 2 cancers-13-00396-f002:**
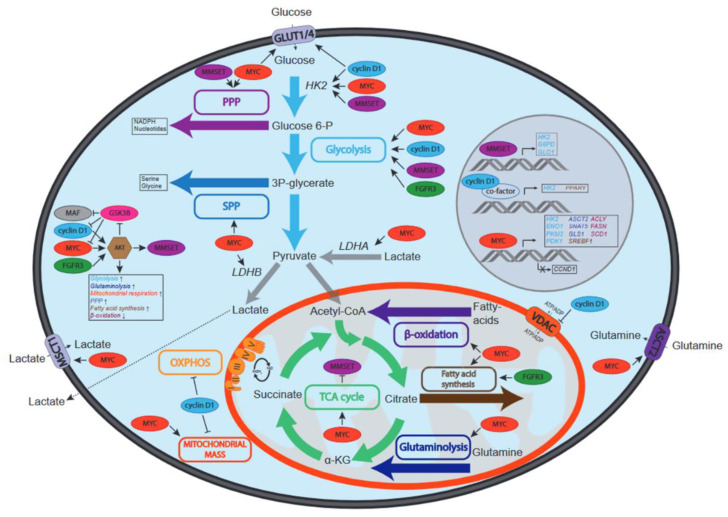
Schematic overview of the effects of recurrent genetic aberrations on metabolic pathways in MM. Depicted are MYC, cyclin D1, MMSET, FGFR3 and MAF and the effects they have on the activity of metabolic pathways (depicted by colored boxes and arrows) and/or metabolite transporters (MSCT1, VDAC, ASCT2, GLUT1 and GLUT4). Increased or decreased activation can be due to regulation of the activity, protein stability or expression of enzymes involved in specified pathways. In the middle left the effects of MYC, cyclin D1, MMSET, FGFR3 and MAF on AKT, a central regulator of metabolism, are depicted. Due to the key roles HK2, LDHA and LDHB play in glucose and lactate metabolism, the effects of oncogenes on these enzymes are also shown.

**Figure 3 cancers-13-00396-f003:**
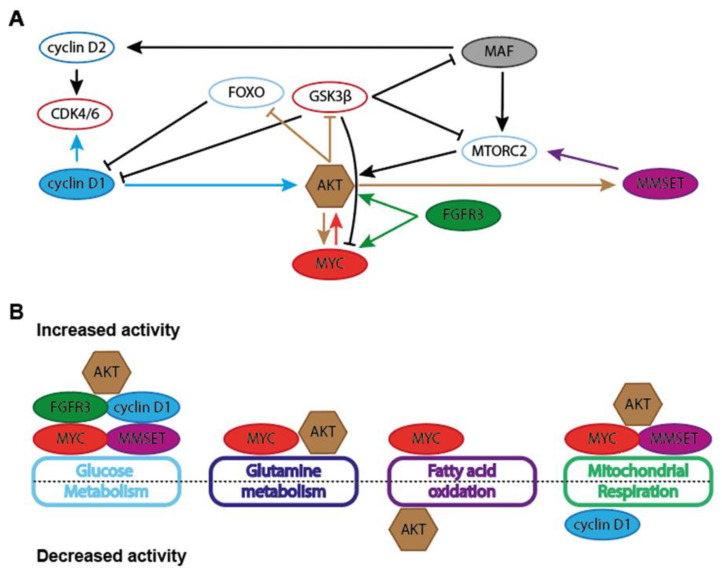
Schematic representation of general metabolic effects of MM-associated oncogenes. (**A**) Interdependencies and functional associations between MYC, cyclin D1, MMSET, FGFR3, AKT and MAF in the regulation of metabolic features in cancer cells. (**B**) Schematic depiction of general effect MM-asociated oncogenes on the activity of several metabolic pathways. Glucose metabolism entails glycolysis, pentose phosphate pathway (PPP), serine synthesis pathway (SSP) and hexosamine biosynthesis pathway (HBP). Mitochondrial respiration includes the tricarboxylic acid (TCA) cycle and oxidative phosphorylation (OXPHOS).

## Data Availability

No new data were created or analyzed in this study. Data sharing is not applicable to this article.

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
