# Peer review of "Metabolic Effects of Recurrent Genetic Aberrations in Multiple Myeloma"

_cancers, 2021, doi:10.3390/cancers13030396_

Round 1

Reviewer 1 Report

This review article is providing both basic researchers and physicians comprehensive information about metabolic effects of chromosomal translocation specific to multiple myeloma to establish novel therapeutic strategies.

1) Twenty percent of patients with multiple myeloma and 80% of myeloma cell lines harbor multiple IGH translocations that involve more than two set of partner chromosomal loci, for example MYC and FGFR3, MYC and CCND1, and FGFR3 and CCND1. What are metabolic effects in these cases?

2) It is desirable to have a couple of tables for readers to easily understand the difference of the metabolic consequences in each recurrent chromosomal translocation.

3) The authors emphasize activation of the AKT pathway in MM cells in “Discussion” section. However, AKT inhibitor, perifosine, demonstrated no clinical effect for MM in the previous clinical trials.

The characters of title should be arranged in sub-number, either uppercase or lower letters.

Author Response

This review article is providing both basic researchers and physicians comprehensive information about metabolic effects of chromosomal translocation specific to multiple myeloma to establish novel therapeutic strategies.

  • Response: We thank the reviewer for the positive assessment of our manuscript.

1) Twenty percent of patients with multiple myeloma and 80% of myeloma cell lines harbor multiple IGH translocations that involve more than two set of partner chromosomal loci, for example MYC and FGFR3, MYC and CCND1, and FGFR3 and CCND1. What are metabolic effects in these cases?

  • Response: The reviewer raises an interesting point with regards to the metabolic effects of co-occurring translocations in MM. However, this question is difficult to answer as experimental data with regards to this issue is lacking, so we can only speculate. It appears that most translocations have largely overlapping metabolic effects and induce a marked increase in metabolic activity in MM, as shown in Figure 3. The CCND1 translocation is a possible exception, as high CCND1 activity can lead to a reduction of mitochondrial respiration. In general, MYC can be regarded as a dominant factor in altering or amplifying metabolic features of MM plasma cells, and other chromosomal events, or constellations thereof, may indirectly result in the activation of MYC and thereby influence metabolism. We have alluded to this point in the ‘Concluding Remarks and Future Perspective’ section of our revised manuscript (lines 968-971). We deemed it beyond the scope of this review to further speculate on this issue, as it is not sufficiently supported by experimental data in our opinion.

2) It is desirable to have a couple of tables for readers to easily understand the difference of the metabolic consequences in each recurrent chromosomal translocation.

  • Response: We thank the reviewer for this helpful suggestion and we have added a table to the manuscript that summarizes the metabolic consequences of the discussed translocations. We have placed the table at the end of the manuscript (p. 45).

3) The authors emphasize activation of the AKT pathway in MM cells in “Discussion” section. However, AKT inhibitor, perifosine, demonstrated no clinical effect for MM in the previous clinical trials.

  • Response: We agree with the reviewer that the clinical efficacy of the AKT inhibitor perifosine in MM patients is dissapointing. However, we are of the opinion that this is no direct evidence against the notion that AKT is a central metabolic node in MM. In the discussion section we draw attention to the fact that constitutive activation of AKT lowers the cellular checkpoint that protects against oncogenic stress, such as provoked by MYC activation for instance. Furthermore, the metabolic effects of AKT inhibition are possibly uncoupled from direct cytotoxic effects. Lastly, we point out that although AKT appears to be of importance for key metabolic features of MM cells, we do not claim that AKT therefore is a superior clinical target. In this review we have largely refrained from discussing drug-associated metabolic consequences in MM cells, as this lies beyond the scope and is the subject of several excellent other reviews that have been published.

The characters of title should be arranged in sub-number, either uppercase or lower letters.

  • Response: We have changed the titles to lower letters in the revised manuscript.

Reviewer 2 Report

This review is well written and comprehensive enough.

I think that this article is ready for publication.

Author Response

This review is well written and comprehensive enough. I think that this article is ready for publication.

  • Response: We thank the reviewer for the positive assessment of our work.

Reviewer 3 Report

The manuscript by Bloedjes et al. compared the metabolic characteristics of multiple myeloma cells to those of normal plasma cells. Further, the authors comprehensively described metabolic dysregulation induced by different oncogenes that are aberrantly expressed in multiple myeloma as a result of various chromosomal translocations. A comprehensive overview of the different metabolic pathways dysregulated by each of these oncogenes was presented and discussed in the context of multiple myeloma pathogenesis. Based on previous reports on how each of these oncogenes controls metabolism in different cell types, the authors derived reasonable speculations on the role of these genes in altering multiple myeloma cell metabolism, therefore providing a framework for understanding metabolic changes in multiple myeloma. Overall, the manuscript is well written and provides extensive knowledge for researchers in the field. Only few sentences in the manuscript are complex and needs to be rephrased to improve their clarity.

Minor comments:

  1. MMSET has been given the HUGO name NSD2 and this is preferred in the literature now
  2. In line 73, the authors need to elaborate a little more on the role of AID in somatic hypermutation and class switch recombination of immunoglobulin genes.
  3. In line 88, the sentence “Using mTORC1…in mice” should be improved
  4. In line 111, the authors suggested that the difference in metabolism between long-lived and short-lived plasma cells are not due to transcriptional changes. Can the authors speculate, based on previous reports, on the molecular changes underlying the metabolic differences? Post-transcriptional/post translational? Signaling?
  5. In line 180, the sentence should be “…in hematological cancers, MYC dysregulation is often a result of chromosomal translocations.
  6. In line 195, please explain the expression “involves duplications and multiple chromosomes”. More information needed here.
  7. In line 344, please rephrase the sentence “ Ammonia is…it was ammonia”. It’s not clear!
  8. In line 366, the sentence: “Based on these …Myc deregulation” is too long. Please break to enhance clarity.
  9. In line 708, please clarify “…leading to a global halt…recovery and survival”. It’s not clear what’s meant here.
  10. In line 723, the authors stated that H3K36me2 by NSD2 results in a more closed chromatin!!
  11. In line 848, please expand “HBP “ at first mention.
  12. The paragraph “line 900-922” is a repetition of the previous paragraph
  13. In line 951, I’m not sure how the first paragraph of this section is related to lipid metabolism!

Author Response

The manuscript by Bloedjes et al. compared the metabolic characteristics of multiple myeloma cells to those of normal plasma cells. Further, the authors comprehensively described metabolic dysregulation induced by different oncogenes that are aberrantly expressed in multiple myeloma as a result of various chromosomal translocations. A comprehensive overview of the different metabolic pathways dysregulated by each of these oncogenes was presented and discussed in the context of multiple myeloma pathogenesis. Based on previous reports on how each of these oncogenes controls metabolism in different cell types, the authors derived reasonable speculations on the role of these genes in altering multiple myeloma cell metabolism, therefore providing a framework for understanding metabolic changes in multiple myeloma. Overall, the manuscript is well written and provides extensive knowledge for researchers in the field. Only few sentences in the manuscript are complex and needs to be rephrased to improve their clarity.

  • Response: We are grateful to the reviewer for the positive feedback on our manuscript.

Minor comments:

MMSET has been given the HUGO name NSD2 and this is preferred in the literature now.

  • Response: Although we agree with the reviewer that the official HUGO nomenclature for this gene is NSD2 we would rather use the name MMSET in this regard, as it appears to be the preferred name for this gene in the multiple myeloma research field. As an example, the combination of the search terms ‘NSD2 AND myeloma’ yields 71 hits in Pubmed, whereas ‘MMSET AND myeloma’ yields 136 hits. However, if the reviewer insists, we will comply with this request and change it to NSD2 throughout the text.

In line 73, the authors need to elaborate a little more on the role of AID in somatic hypermutation and class switch recombination of immunoglobulin genes.

  • Response: We thank the reviewer for this suggestion and have added additional information on the roleof AID in somatic hypermutation and class switch recombination of immunoglobulin genes in this section (lines 74-84). Also, we briefly alluded to the presumed role of AID in the generation of driver mutations and recurrent chromosomal translocations in multiple myeloma.

In line 88, the sentence “Using mTORC1…in mice” should be improved

  • Response: We agree and have rephrased this sentence to: “Rapamycin treatment abrogated plasma cell differentiation in mice, indicating that mTORC1 is of crucial importance for the generation of plasma cells.” (lines 97-98).

In line 111, the authors suggested that the difference in metabolism between long-lived and short-lived plasma cells are not due to transcriptional changes. Can the authors speculate, based on previous reports, on the molecular changes underlying the metabolic differences? Post-transcriptional/post translational? Signaling?

  • Response: In the referenced paper (Lam et al. Cell Rep. 2018, PMID: 30157439), the authors show that uptake of glucose correlated with long halve-lives in plasma cell subsets. It was shown that plasma cells with diminished glucose uptake maintained translation rates but secreted relatively few antibodies when compared to plasma cells with higher glucose uptake. Previous work of these authors showed that imported glucose was used mainly for glycosylation of antibodies and to provide spare respiratory capacity to plasma cells. Long-lived plasma cells also express high levels of CD98, a common subunit of several amino acid transporters and this was coupled to increased autophagy activity. Furthermore, the authors report that glutamine was used almost exclusively for anaplerotic reactions to generate glutamate and aspartate, which also provides electrons for respiration through succinate oxidation. Amino acid concentrations were also a limiting factor for antibody secretion. Based on these observations we speculate that the difference in metabolism between long-lived and short-lived plasma cells are related to subtle transcriptional changes associated with the plasma cell differentiation continuum (such as CD98 expression), and differential signaling through, for instance, receptor tyrosine kinases, dependent on niche variation and nutrient availability. However, we would like to stress that the exact causes for the observed differences remains unclear. We have added a sentence on this aspect to this section (lines 122-127).

In line 180, the sentence should be “…in hematological cancers, MYC dysregulation is often a result of chromosomal translocations.

  • Response: We concur fully with this rephrasing and have changed the sentence in accordance with the suggestion of the reviewer (lines 194-195).

In line 195, please explain the expression “involves duplications and multiple chromosomes”. More information needed here.

  • Response: To clarify this issue, we have changed this sentence to: “The IGH MYC rearrangements in MM are characterized by chromosomal duplications of the breakpoint region and often involve other genes in addition to IGH and MYC.” (lines 208-210).

In line 344, please rephrase the sentence “ Ammonia is…it was ammonia”. It’s not clear!

  • Response: We apologize for the unclear phrasing. We have changed this sentence to: “Ammonia is the toxic byproduct of glutamine breakdown. Conversely, ammonia is consumed during glutamine synthesis.” (lines 358-359).

 In line 366, the sentence: “Based on these …Myc deregulation” is too long. Please break to enhance clarity.

  • Response: We have shortened this section and broken it into several sentences to improve clarity. The section was changed to: “Based on these observations we speculate that elevated 2-HG in MM may be part of a positive feedback mechanism. Oncogenic MYC drives the expression of 2-HG, which results in mTORT/AKT-mediated downmodulation of FOXO that enables further MYC deregulation.” (lines 380-383).

In line 708, please clarify “…leading to a global halt…recovery and survival”. It’s not clear what’s meant here.

  • Response: We agree with the reviewer and have changed this part of the manuscript as follows: Was changed to: “Activation of this response resulted in phosphorylation of the eukaryotic translation initiation factor 2a (eIF2a) and subsequent attenuation of global mRNA translation. Simultaneously, phosphorylation of eIF2a allows for the selective translation of proteins involved in cell survival and recovery, such as ATF4.” (lines 721-724).

In line 723, the authors stated that H3K36me2 by NSD2 results in a more closed chromatin!!

  • Response: We thank the reviewer for bringing this error to our attention. In this sentence we changed “closed” to “open”, as H3K36Me2 enhances transcription. We also noted that the two references in this sentence were incorrect due to editing errors. We have changed these accordingly. (line 738).

In line 848, please expand “HBP“ at first mention.

  • Response: In the manuscript, we refer to the HBP (hexosamine biosynthesis pathway) for the first time in line 129, where it is expanded at first mention.

The paragraph “line 900-922” is a repetition of the previous paragraph

  • Response: We thank the reviewer for notifying us of this duplication error. We have deleted this duplication in the revised version.

In line 951, I’m not sure how the first paragraph of this section is related to lipid metabolism!

  • Response: We agree with the reviewer that this part is not related to the role of MAF in lipid metabolism. We have deleted the first paragraph of this section in the revised version.